# Performance and Participation in the ‘Vasaloppet’ Cross-Country Skiing Race during a Century

**DOI:** 10.3390/sports7040086

**Published:** 2019-04-12

**Authors:** Nastja Romancuk, Pantelis T. Nikolaidis, Elias Villiger, Hamdi Chtourou, Thomas Rosemann, Beat Knechtle

**Affiliations:** 1Institute of Primary Care, University of Zurich, Zurich 8006, Switzerland; nastjaromancuk@gmail.com (N.R.); evilliger@gmail.com (E.V.); thomas.rosemann@usz.ch (T.R.); 2Exercise Physiology Laboratory, Nikaia 18450, Greece; pademil@hotmail.com; 3Activité Physique: Sport et Santé, UR18JS01, Observatoire National du Sport, Tunis 2020, Tunisie; h_chtourou@yahoo.fr; 4Institut Supérieur du Sport et de l’éducation physique de Sfax, Université de Sfax, Sfax 3000, Tunisie; 5Medbase St. Gallen Am Vadianplatz, St. Gallen 9001, Switzerland

**Keywords:** aerobic capacity, cross-country skiing, nationality, participation, performance, sex difference, ultra-endurance

## Abstract

This study investigated gender differences in performance and participation and the role of nationality during one century in one of the largest cross-country (XC) skiing events in the world, the ‘Vasaloppet‘ in Sweden. The total number of female and male athletes who finished (n = 562,413) this race between 1922 and 2017 was considered. Most of the finishers were Swedish (81.03% of women and 88.39% of men), followed by Norwegians and Finnish. The overall men-to-women ratio was 17.5. A gender × nationality association was observed for participation (χ^2^ = 1,823.44, p < 0.001, φ = 0.057), with the men-to-women ratio ranging from 6.7 (USA) to 19.1 (Sweden). For both genders, the participation (%) of Swedish decreased, and that of all other nationalities (except Swiss) increased across years. Regarding the mean race time, men were faster than women by 14.5% (7 h 52 min 17 s versus 9 h 00 min 55 s, respectively). A trivial gender×nationality interaction regarding the race time was observed (p < 0.001, η^2^ < 0.001), with gender differences ranging from 4.4% (USA) to 22.0% (Iceland). The race time increased across calendar years for both women (r = 0.45, p = 0.006, moderate magnitude) and men (r = 0.25, p = 0.015, small magnitude). On the basis of these findings, we concluded that a relatively small number of women pariticipates in XC skiing. Therefore, the development of public health policies targeting the participation of women in XC skiing should be a concern in the countries with a tradition of this sport.

## 1. Introduction

The ‘Vasaloppet’ is one of the most popular cross-country (XC) skiing races worldwide and among the oldest outdoor sport events held since 1922 [1]. XC skiers competing in this race cover the distance of 90 km using the classical XC skiing technique [2]. The pacing is variable and depends on gender, age, and performance level; women, young, and fast XC skiers show relatively smaller changes in speed during the race than other participants [2,3]. The physiological aspects of this race have been well studied for half a century [4,5]. More recently, this race became the subject of studies on the relationship between exercise and mortality [6,7], incidence of injuries [8], cardiovascular complications [9,10], and cancer [11]. In contrast to the extensive studies on physiological and pathophysiological aspects of this race, limited researches have been conducted with regards to trends in participation and performance across years [2,12].

There is little knowledge about the female and male participation in this race in general and especially across years. There are not many studies investigating the gender differences in performance in XC sking on the basis of a large enough and representative number of participants. There is a study regarding male/female differences on the impact of exercise intensity on upper-body poling, but it was conducted in eight male and eight female elite skiers [13]. Another study investigates the influence of poling on endurance performance in eight male and eight female elite XC skiers [14]. The men-to-women ratio has been previously used as an index of the variation of participation by gender. A study of the ‘Vasaloppet’ during a short period (2012–2016) revealed an association between calendar year and gender participation, with the men-to-women ratio being the smallest in 2015 and the highest in 2012 [2], indicating a trend of increased participation of women. The smallest men-to-women ratio was observed in the under-21-years age group and the largest one in the 70–74-years age group [2].

Moreover, little information exists with regard to trends of performance in ‘Vasaloppet’ across years. So far, research on differences in performance by gender indicated that women and men achieved similar race times in XC skiing events such as the ‘Engadin Ski Marathon’ held in Switzerland when all finishers were considered, but men were faster by ~7% when only the top 10 finishers were analyzed [15]. In addition to gender and age, another factor that may influence participation and performance is nationality. Indeed, studies on sports such as long-distance running showed differences in performance participation by nationality, and these differences may vary by gender and age [16]. For example, athletes originating from East Africa dominate in long-distance running such as marathon running [16,17], whereas Japanese runners are the fastest worldwide in 100 km ultra-marathons [16]. 

Also, we know from other endurance sports that local athletes dominate. For example, when considering nationality and performance in triathletes competing in ‘Ironman Hawaii’ held in the United States of America, the fastest race times were achieved by U.S. Americans [18], while the fastest triathletes participating the ‘Norseman Xtreme Triathlon’ held in Norway were Norwegian [19]. The fastest triathletes competing between 2002 and 2015 in 253 different Ironman triathlon races carried out globaly, originated from Germany [20]. Most swimmers, who crossed the 33.8 km ’English Channel’, came from Great Britain, the United States of America, Australia, and Ireland, and these nationalities were the fastest, too [21]. One possible explanation for the dominance of local athletes in these sport disciplines might be the historical development and popularity of these sports in the nations where these competitions are being held. 

Very little is known on the aspects of nationality and performance in XC skiers. In XC skiing, only one recent study showed an effect of nationality on performance in the ‘Engadin Ski Marathon’ held in Switzerland, with Russians being the fastest, and Swiss being the most numerous finishers [22]. In other sports such as the ’Norseman Xtreme Triathlon‘ [19] or the ’Ironman Hawaii‘ [18], most of the dominating athletes originated from the country where the sport event was held. Despite the popularity of the ‘Vasaloppet’ and its long history, no study has ever examined the trends of performance and participation across years and their variation by nationality. Considering the relationship between participation in the ‘Vasaloppet’ and cardiovascular health (i.e., the positive effect of exercise on health) [7], knowing the variation of participation and performance across calendar years by nationality is important, not only for coaches and fitness trainers working with XC skiers but also from a health perspective. A large study comparing skiers in this XC race and non-skiers provided additional information for the hypothesis that individuals with a high level of physical activity have a healthy lifestyle and a lower risk of CVD (Cardiovascular Diseases) or death [23]. 

Therefore, the aim of the present study was to examine the trends in participation and performance in the ‘Vasaloppet’ across calendar years with regards to gender and nationality. On the basis of the findings of a recent study on XC skiing in the ’Engadin Ski Marathon‘ [22] and on other competitions such as the ’Norseman Xtreme Triathlon‘ [19] and ’Ironman Hawaii‘ [18], we hypothesized that most of the athletes who finish the race would be local (Swedish), and Swedish athlethes would be the fastest for both men and women.

## 2. Materials and Methods

### 2.1. Participants

The Institutional Review Board of Kanton St. Gallen, Switzerland, approved all procedures of this study with a waiver of the requirement for informed consent of the participants, given the fact that the study involved the analysis of publicly available data. To test our hypothesis, we considered all official finishers, since the first edition of ‘Vasaloppet’ held in 1922. We investigated the effects of nationality, gender, and year on race time. All XC skiers (n = 562,413) who finished the ‘Vasaloppet’ from 1922 to 2017 were considered for the analysis. 

### 2.2. Procedures

Data were obtained from the official race website www.vasaloppet.se/en/. The ‘Vasaloppet’ is a long-distance XC ski race performed in classic technique, annualy held in Sweden at the beginning of March. The course is 90 km long and, therefore, one of the longest XC races in the world. The first competition was held in 1922, which makes it also the oldest XC race [24]. Women were allowed to participate in 1922, 1923, and from 1981 but did not receive awards until 1997 [25]. In 2017, approximately 16,000 participants participated in this massive event [24]. Nationality and gender of the finishers was only considered when it was ≥0.1% of the total sample, resulting in the analysis of 14 nationalities (i.e., Swedish, Norwegian, Finnish, German, Danish, Italian, Czech, Swiss, Estonian, Austrian, French, Russian, U.S. American, and Icelander), whereas nationalities with <0.1% were grouped as ‘Other’.

### 2.3. Statistical Analyses

All statistical analyses were performed using the statistical package IBM SPSS v.20.0 (SPSS, Chicago, IL, USA). The figures were created using the software GraphPad Prism v. 7.0 (GraphPad Software, San Diego, CA, USA). Data are presented as mean ± standard deviation. We examined the association of gender with nationality, i.e., whether the distribution of gender varied by nationality, using chi-square (χ^2^), and Cramer’s phi (φ_C_) evaluated the magnitude of association. Men-to-women ratio, i.e., the number of men divided by the number of women finishers, was calculated for the whole sample and for each nationality. A two-way ANOVA examined the main effects of nationality and year, differences between women and men, and the gender×nationality and gender×year interaction regarding race time and age, followed by a Bonferroni post-hoc analysis. The magnitude of differences in ANOVA was evaluated using eta squared (η^2^) as trivial (η^2^ < 0.01), small (0.01 ≤ η^2^ < 0.06), moderate (0.06 ≤ η^2^ < 0.14), and large (η^2^ ≥ 0.14) [26]. Pearson correlation coefficient (r) examined the relationship between age and race time. The trend of race time across years was examined using linear regression analysis. Gender difference was calculated using the formula 100 × (race time in women − race time in men)/race time in men. Alpha level was set to 0.05. 

## 3. Results

### 3.1. Overall Trends in Participation

More than 9 out of 10 finishers were Swedish, Norwegians, and Finnish (Table 1). The overall men-to-women ratio was 17.5. A gender×nationality association was observed in participation (χ^2^ = 1,823.44, p < 0.001, φ = 0.057), with the men-to-women ratio ranging from 6.7 (USA) to 19.1 (Sweden). The percentage of women was greater for Norwegians and Finnish than for Swedish. The men-to-women ratio for nationalities with <0.1% finishers ranged from 1.3 (Belarus) to 39.3 (Aland Islands) (Table 2). 

### 3.2. Overall Trends in Performance

A gender difference of trivial magnitude was shown (p < 0.001, η^2^ = 0.002), with men being faster than women by 14.5% (7 h 52 min 17 s versus 9 h 00 min 55 s, respectively) (Figure 1). A small main effect of nationality on race time was shown (p < 0.001, η^2^ = 0.010) with Russians, Norwegians, and Austrians being the fastest, and Danish the slowest. A trivial gender×nationality interaction on race time was observed (p < 0.001, η^2^ < 0.001) with gender difference ranging from 4.4% (USA) to 22.0% (Iceland). In women, a small main effect of nationality on race time was found (p < 0.001, η^2^ = 0.049), with Russians, Norwegian, Austrians, and Estonians being the fastest, and Danish and Icelanders the slowest. In men, a small main effect of nationality on race time was found (p < 0.001, η^2^ = 0.039), with Russians, Norwegians, and Austrians being the fastest, and Danish being the slowest. That is, the main effect of nationality (i.e., effect size) was stronger in women than in men.

### 3.3. Trend of Participation over Years

In women, the participation (%) of Swedish decreased, and that of all other nationalities (except Swiss) increased across years (Figure 2). The magnitude of the relationship between participation and calendar year ranged by nationality from moderate (i.e., USA) to very large (i.e., Denmark). Similar trends were observed for men (Figure 3), where the magnitude of the relationship ranged from large (i.e., Czech) to very large (i.e., Finland).

### 3.4. Trend of Performance across Calendar Years

The race time increased across time for both women (r = 0.45, p = 0.006, moderate magnitude) and men (r = 0.25, p = 0.015, small magnitude) (Figure 4). In women, the race time increased for Swedish, Russian, and Icelander finishers, decreased for Swiss finishers, whereas it remained stable for the other nationalities (Figure 5). In men, it increased for Swedish, Finnish, Estonian, French, Icelander, and Other, decreased for Norwegian, Italian, Austrian, Russian. and USA finishers, whereas it did not change for the other nationalities (Figure 6).

## 4. Discussion

The main findings of the present study are that: (1) over 80% of the finishers in the ‘Vasaloppet’ were Swedish, (2) the participation increased over the years overall and for both genders, (3) the race time increased, especially for women, (4) the Russians were the fastest participants overall and for both genders. 

### 4.1. Overall Trends in Participation

A first important finding is that over 80% of the participants in ‘Vasaloppet’ originated from the country where the race takes place, which is in agreement with previous studies regarding running, cycling, and swimming events, such as the ‘Norseman Xtreme Triathlon’ from 2006 to 2014 [12], the ‘Ironman Hawaii’ from 1985 to 2012 [16], and the ‘English Channel Swim’ from 1875 to 2013 [18,19,21] that show a similar trend. It is noteworthy that until the 1990s, almost all finishers were Swedish. This period can be considered as a landmark; thereafter, the percentage of Swedish finishers has decreased continuously, and that of other nationalities has increased. The entrance of Sweden in the European Union in 1995 might have facilitated the participation of foreigners, especially those from countries of the European Union. The overall increased participation across calendar years should be attributed to a general trend observed in endurance and ultra-endurance events over the past few decades [27]. In addition, geographical proximity plays an important role in the participation, as a large proportion of finishers came from the other two Scandinavian countries (i.e., Norway and Denmark), Finland, and Germany. Countries with a tradition of this sport, such as Central European countries, also had an important presence in the race. 

We observed that nationality influenced the rates of participation and the men-to-women ratios. Surprisingly, the local finishers presented one of the largest men-to-women ratio, which was in contrast with previous research in other XC races, such as the ‘Engadin Ski Marathon’ [22], showing that local finishers included relatively more women than finishers of other nationalities. Accordingly, we would expect more local women participating compared to foreigners because of gender differences regarding barriers to exercise (e.g., relative difficulty of traveling for women). A possible explanation might be differences concerning facilitations for young athletes in Sweden and Switzerland or the number of XC skiing clubs. Regarding other XC races such as the ‘Engadin Ski Marathon’ held in Switzerland, the sport clubs in Switzerland manage to reach and inspire more young people to perform this sport and thus participate in XC skiing competitions like the ‘Engadin Ski Marathon’, since this sport has gained a lot of popularity thanks to the success of elite XC ski athletes such as Dario Cologna [28]. However, further research is needed to confirm this potential explanation. 

### 4.2. Trend of Participation across Calendar Years

The overall number of female participants increased for all nationalities over the years, but when looking at nationalities separately, the number of participants decreased for Swiss and Swedish athletes. In men, the magnitude of the relationship between participation and calendar year according to nationality ranged from large (i.e., Czech) to very large (i.e., Finland). The increased participation of athletes in this event might be attributed to a general growing popularity of XC skiing, a trend that has been observed for other events, such as the ‘Engadin Ski Marathon’ [22]. Most participants were from Sweden, followed by Norway and Finland, which might be explained by the popularity of XC skiing in these countries and also by the fact that Norway and Finland are neighbouring countries. These findings were reported for other sport events such as the ‘Norseman Xtreme Triathlon’, where most of the participants originated from Norway [19], and ‘Ironman Hawaii’, where most athletes were American [18]. 

Furthermore, a year-to-year variation of participation could be attributed to environmental conditions. For instance, a study of the numbers of registered and starting skiers in ‘Vasaloppet’ for the period 1951–2016 showed that a lack of natural snow significantly increased the cancellation ratio; thus, a decline in snow depth from the average level of 57 cm to 30 cm was observed, resulting in an increase of 3.3% percentage of the cancellation rate during this period [12]. However, the environmental conditions do not seem to affect participation in the long term. The particular variation of snow depth nowadays can be offset by practices such as snow storage and snow production [12].

### 4.3. Trend of Performance across Calendar Years

The regression analysis of race time by calendar year showed that the race time increased, and this trend was stronger for all women than for all men who participated from 1922 until 2017. This trend could be attributed to the increased rates of participation across calendar years, that is, the race has become a “massive” event attracting more recreational participants, and, consequently, the more the participants, the slower the race time. Accordingly, this trend was stronger for women than for men, because the participation of women has been increasing more than that of men. Also, as found in this study, the Russians were the fastest participants for both genders but, considering the total number of participants, they only represented 0.22% of women and 0.14% of men. 

The Russians were also the fastest athletes in other sport events such as the ‘Engadin Ski Marathon’ in Switzerland [22] or the 100 km ultra-marathon races held worldwide [16]. One explanation for this success might be that winter sports are very popular in Russia, since they are well supported by the Sports State Committee. A reason for the State support might be the prestigious function of the Russian athletes in representing their country at sport events [29,30]. The Soviet Union government’s sport institutions, such as Vsevobuch, Red Star International, and All-Union Sports Committee, used to support dedicated athletes, who would represent political autonomy, military readiness, and athletic dominance [31]. Also, the former Soviet Union supported Russian army physicians in conducting extensive research on improving the fitness of soldiers and enhancing their acclimatization to different environments [32,33]. 

Nevertheless, it is noticable that Russian athletes dominate in sports that are not as popular in Russia. The increasing success of a nation in different sports raises questions about a possible artificial increase in performance. In 2016, the WADA’s (World Anti-Doping Agency) independent McLaren Investigation verified accusations of manipulation of the doping control process before, during, and after the 2014 Sochi Olympic and Paralympic Games [34]. Already in 2010, a doping scandal during the olympics occurred, involving biathletes from Russia [35]. 

These past scandals and current blood-doping scandals involving five athletes at the Nordic World Ski Champoionships held in Seefeld, Austria [31], show that the abuse of drugs and the use of questionable methods to enhance the physical performance are a problem in XC skiing events. Regarding, specifically, XC races, we are not able to confirm the possibility of an abuse of performance-enhancing drugs, since these event have existed for decades, and some of the races started before official testing for prohibited substances were available. Testing for prohibited substances was primarily introduced for very large events such as the Olympic Games or World Championships [36].

### 4.4. Strength and Limitations

The strength of this study is the sample size (∼560,000 finishers), collected form the first performance in 1922 to 2017, which allowed a comparison of participation and performance for men and women. The findings of the present study must be carefully compared to findings from research on other races, because of the influence of the country where the race is held on XC skiers’ participation. In the ’Vasaloppet‘ held between 1922 and 2017, most finishers were Swedish for both genders, but athletes from Russia were the fastest for both men and women. One explanation for this finding might be the support of the Russian Sports State Committee to XC skiing associations. Further studies are needed to investigate other XC skiing races to determine from which nation the fastest XC skiers originate and if there are differences regarding gender for the participating nations. Because of the growing popularity of XC skiing in countries of the northern hemisphere, these findings interest a large audience. These findings might help strength and conditioning coaches to understand the trends in performance and participation in such grand sport events. The findings of the present study can be applied by strength and conditioning coaches and athletes to improve their training and competition performance. For instance, knoweldge of differences in performance among nationalities might help XC skiers during the race and might be indicative of the performance of their opponents. Athletes and strength and conditioning coaches should be aware that athletes from specific nations may dominate in certain sports disciplines, independent from the location of the competitions. This might be due to their specific commitment to a certain sport discipline.

## 5. Conclusions

On the basis of these findings, we concluded that a relatively small number of women participates in XC skiing. Nevertheless, it is noticable that, over the years, the participation of women in XC skiing has increased, which might lead to a change of the men-to-women ratio in the future. The increased participation of women has already an impact on their race time. Therefore, the development of public health policies targeting the participation of women in XC skiing should be a concern in countries with a tradition of this sport.

## Figures and Tables

**Figure 1 sports-07-00086-f001:**
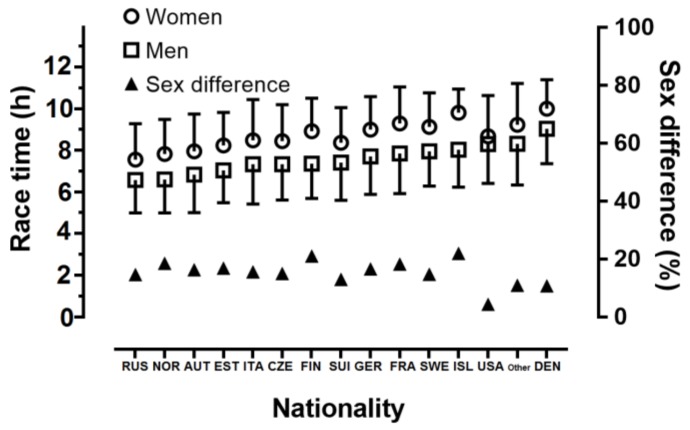
Performance by gender and nationality.

**Figure 2 sports-07-00086-f002:**
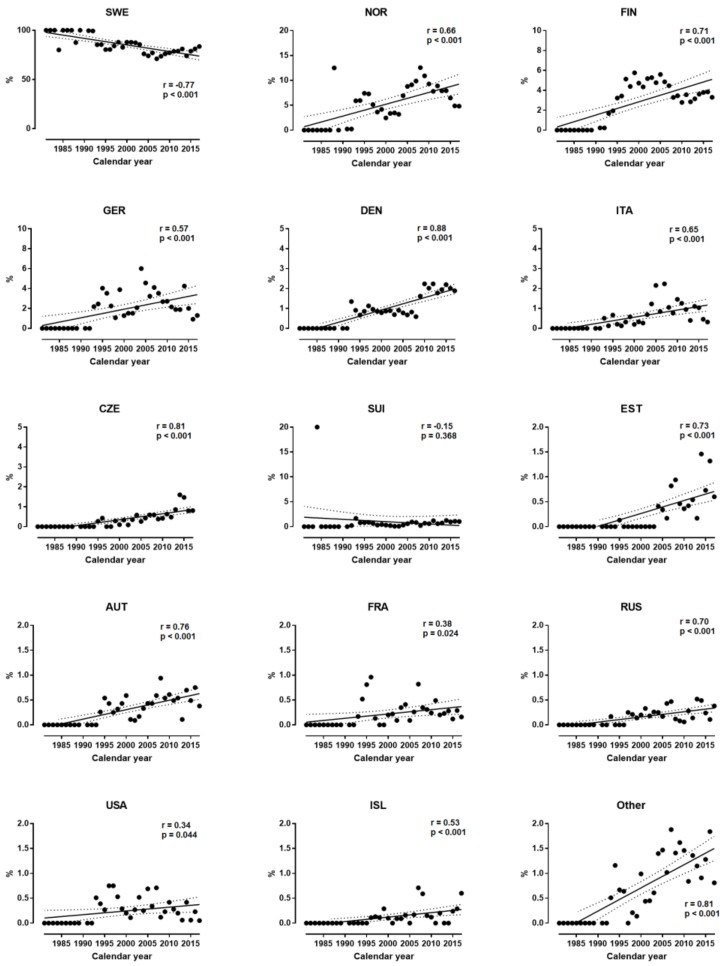
Participation of women by nationality across years.

**Figure 3 sports-07-00086-f003:**
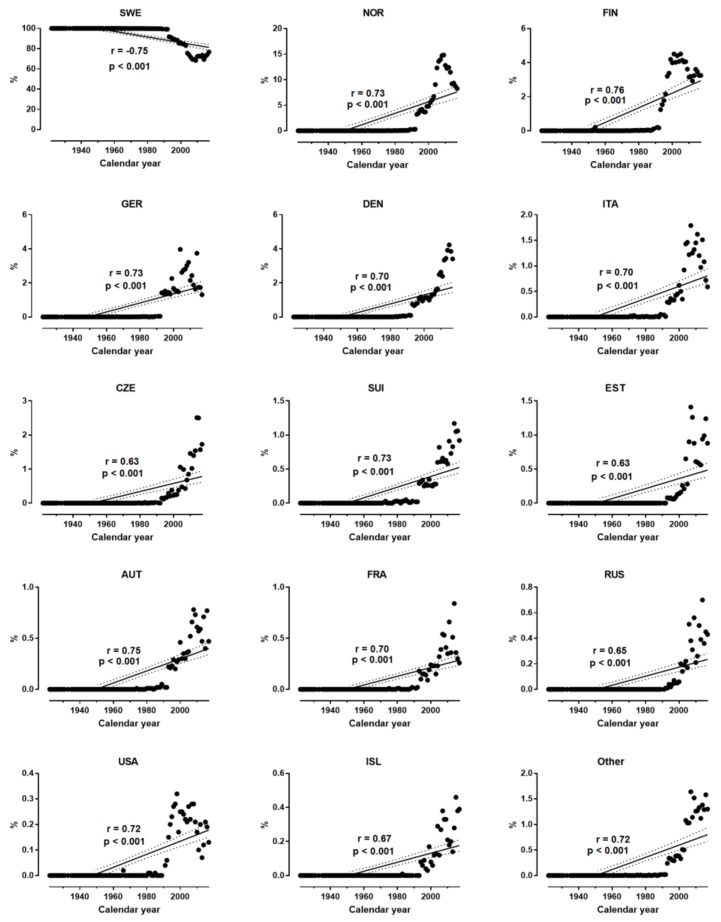
Participation of men by nationality across years.

**Figure 4 sports-07-00086-f004:**
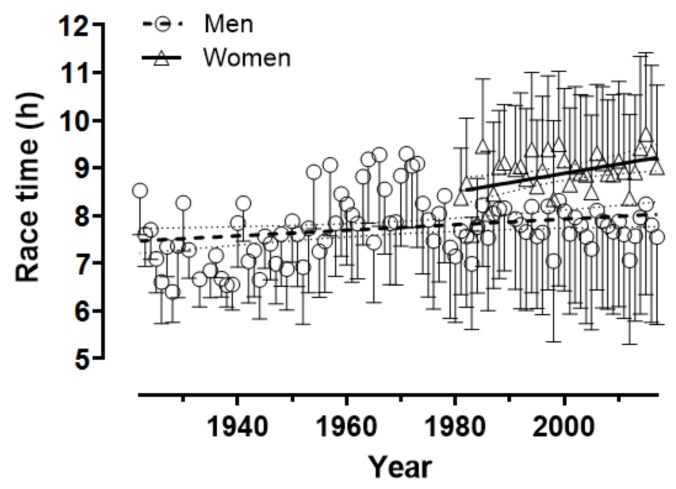
Performance across years.

**Figure 5 sports-07-00086-f005:**
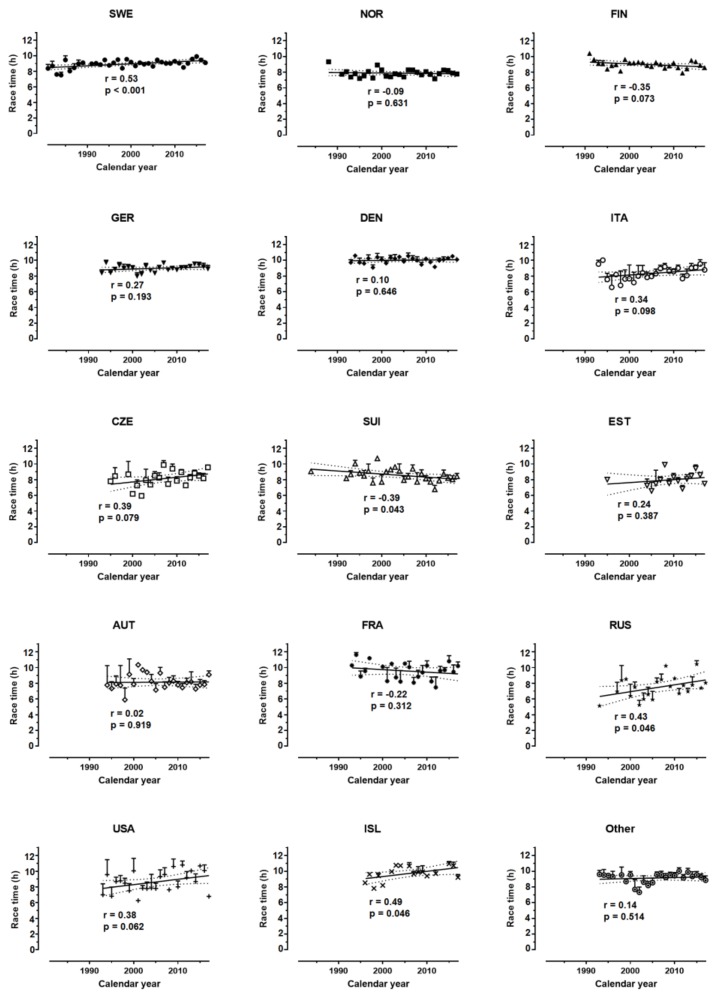
Performance across years by nationality in women.

**Figure 6 sports-07-00086-f006:**
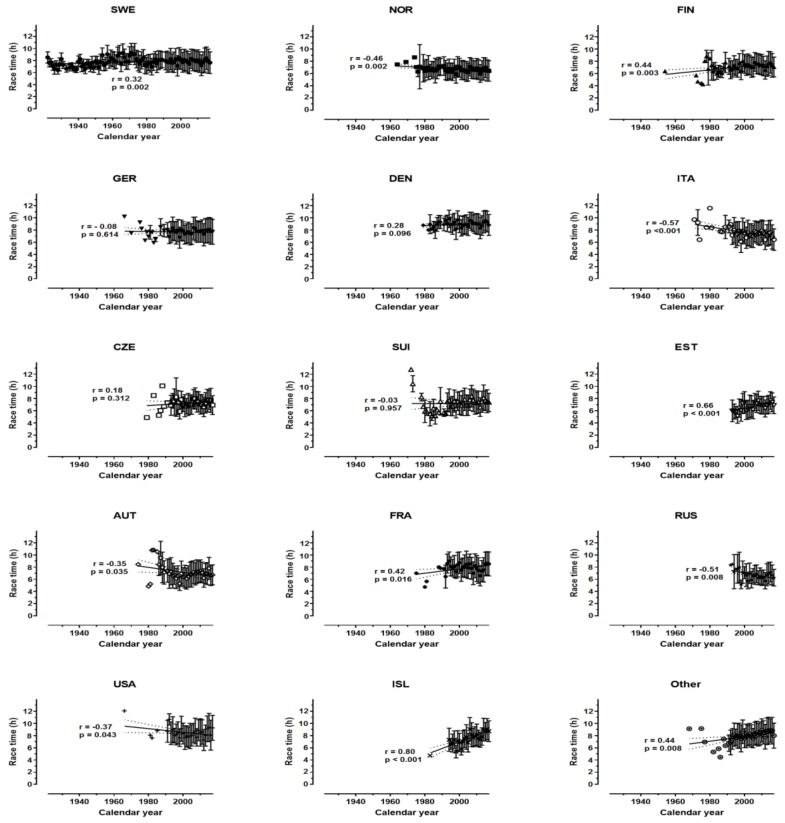
Performance across years by nationality in men.

**Table 1 sports-07-00086-t001:** Participants and men-to-women ratios by nationality.

Nationality	Women		Men		Total		Men-to-Women Ratio
-	n	%	n	%	n	%	-
SWE	24,583	81.03	470,281	88.39	494,864	87.99	19.1
NOR	1997	6.58	25,044	4.71	27,041	4.81	12.5
FIN	1139	3.75	9994	1.88	11,133	1.98	8.8
GER	753	2.48	6084	1.14	6837	1.22	8.1
DEN	411	1.35	5582	1.05	5993	1.07	13.6
ITA	233	0.77	2653	0.50	2886	0.51	11.4
CZE	161	0.53	2460	0.46	2621	0.47	15.3
SUI	211	0.70	1706	0.32	1917	0.34	8.1
EST	123	0.41	1519	0.29	1642	0.29	12.3
AUT	127	0.42	1318	0.25	1445	0.26	10.4
FRA	85	0.28	928	0.17	1013	0.18	10.9
RUS	68	0.22	742	0.14	810	0.14	10.9
USA	92	0.30	617	0.12	709	0.13	6.7
ISL	50	0.16	560	0.11	610	0.11	11.2
Other	304	1.00	2588	0.49	2892	0.51	8.5
Total	30,337	100.00	532,076	100.00	562,413	100.00	17.5

**Table 2 sports-07-00086-t002:** Finishers by gender and men-to-women ratios for nationalities with <0.1% finishers.

Nationality	Women	Men	Total	Men-to-Women Ratio
ALA	41	508	549	12.4
NED	59	443	502	7.5
CAN	51	197	248	3.9
GBR	21	199	220	9.5
SVK	10	180	190	18.0
ESP	21	139	160	6.6
SLO	19	133	152	7.0
POL	8	123	131	15.4
LAT	3	118	121	39.3
Missing information	11	87	98	7.9
AUS	15	62	77	4.1
JPN	10	51	61	5.1
BEL	3	44	47	14.7
LUX	2	44	46	22.0
LIE	4	28	32	7.0
ISR	1	27	28	27.0
HUN	8	19	27	2.4
NZL	3	22	25	7.3
RSA	2	21	23	10.5
IRL	1	19	20	19.0
GRL	4	12	16	3.0
GRE	0	14	14	-
LTU	1	9	10	9.0
PAK	0	10	10	-
CHI	0	8	8	-
CHN	2	6	8	3.0
BLR	3	4	7	1.3
AND	0	6	6	-
POR	0	5	5	-
IND	0	4	4	-
KAZ	0	4	4	-
UKR	0	4	4	-
BRA	0	3	3	-
BUL	0	3	3	-
ENG	0	3	3	-
MEX	0	3	3	-
ROU	0	3	3	-
VEN	0	3	3	-
IRI	0	2	2	-
SIN	0	2	2	-
UAE	0	2	2	-
AFG	0	1	1	-
ARG	0	1	1	-
COL	0	1	1	-
CRC	0	1	1	-
ECU	0	1	1	-
ESA	0	1	1	-
GEO	0	1	1	-
GGY	0	1	1	-
HKG	0	1	1	-
JAM	0	1	1	-
MLT	0	1	1	-
SWZ	0	1	1	-
THA	1	0	1	0.0
UGA	0	1	1	-
UMI	0	1	1	-

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
