# Peer review of "Performance and Participation in the ‘Vasaloppet’ Cross-Country Skiing Race during a Century"

_sports, 2019, doi:10.3390/sports7040086_

Round 1
Reviewer 1 Report
The large data is impressive and makes the results worthwhile. For the reason of the large data set this study becomes publishable. The research is not original but few studies use such a large sample size.
In terms of changes;
a) the authors need to make stronger case for doing the gender analysis; a case they should make is that studies that compare male/female differences rarely use large enough and representative samples.
b) The authors should describe the results in terms of differences between males and females rather than the effects of gender.
c) Speculative comments in the discussion should be removed.
Author Response
Comments and Suggestions for Authors
The large data is impressive and makes the results worthwhile. For the reason of the large data set this study becomes publishable. The research is not original but few studies use such a large sample size.
In terms of changes;
The authors need to make stronger case for doing the gender analysis; a case they should make is that studies that compare male/female differences rarely use large enough and representative samples.
Answer: We agree with the expert reviewer and changed line 46-53 to: There are not many studies, having a large enough and representative number of participants to investigate the differences in performance in genders in cross country sking. There are studies regarding male/female differences on the impact of exercise intensity on gender differences in upper-body poling, but only in eight male and eight female elite skiers (A. M. Hegge PLoS One. 2015; 10(5): e0127509. https://www.ncbi.nlm.nih.gov/pmc/articles/PMC4441444/). Other studies investigate the influence of poling on gender differences in endurance performance in eight male and eight female elite cross-country skiers (Sandbakk Ø; Scand J Med Sci Sports. 2014 Feb;24(1):28-33. https://www.ncbi.nlm.nih.gov/pubmed/22621157).
The authors should describe the results in terms of differences between males and females rather than the effects of gender.
Answer: We agree with the expert reviewer and corrected this aspect throughout the text. E.g. in l.123 “differences between women and men,”, l.141, “A trivial main effect of sex on race time was shown” changed to “A sex difference of trivial magnitude was shown”.
Speculative comments in the discussion should be removed.
Answer: We agree with the expert reviewer and we removed speculative comments and added research that supports certain findings with: Line 212-217: ...young people to perform this sport and thus participate at XC skiing competitions like the ‘Engadin Ski marathon’ since this sport gained a lot of popularity because of the success of elite cross country ski athletes such as Dario Cologna (Andreas Kopp, Davos 11.12.2016, NZZ: Im Schlepptau Dario Colognas https://www.nzz.ch/sport/skisport/langlauf-im-schlepptau-dario-colognas-ld.133913). However, further research is needed to confirm this potential explanation.
Line 252-258: Soviet Union’s government’s sport institution such as Vsevobuch, Red Star International and All-Union Sports Committee, would supported dedicated athletes, who would represent political autonomy, military readiness and athletic dominance (http://blogs.bu.edu/guidedhistory/russia-and-its-empires/ian-blau/).
Also the former Soviet Union supportet russian army physicians on conduction extensive research on improving the fitness of soldiers and enhancing their acclimatization to different envoirements. (Gippenreiter E, West JB. 1996; Riordan J. 1987).
Line 266-274: These past scandals and current blood-doping scandals involving 5 athletes at the Nordic World Ski Champoionships held in Seefeld, Austria, (https://www.bbc.com/news/world-europe-47415803) show us, that the abuse of drugs and questinable methods to enhance physical performances are a problem in cross country skiing events. Regarding specificly the XC reaces we are not able to confirm, the possibility for an abuse of performance enhancing drugs, since this event exists since decades and in some of the races started before official testing for prohibited substances. Testing for prohibited substances was primarily introduced for very large events such as Olympic Games or World Championships (Fraser AD. Doping control from a global and national perspective. Ther Drug Monit. 2004 Apr;26(2):171-4.).
Reviewer 2 Report
This paper examines performance and participation in the Vasaloppet XC skiing race over almost a century. There are few previous publications on this subject, and those that exist are cited. The introduction and methods are appropriate and detailed enough to convey an understanding of the project. I have a couple of suggestions for grammer/spelling, which are minor, but also a concern with the relation to health. All of these should be easily addressed and I have detailed them as follows:
Line 128 Finish should be corrected to Finnish
Line 156 Finish should be corrected to Finnish
Lines 195 and 203 'Engadiner ski marathon' should be consistently changed to 'Engadin ski marathon'
Line 240 rises should be corrected to raises
Line 244 biathlets should be corrected to biathletes and russia should be Russia
Lines 79-83 suggest are a strong follow up statements supporting the logic of the investigation. However, given the results and discussion, very little attention was given to importance from a health perspective. I understand public policy and appreciate the attempt at acknowledging the lack of participation of females, but would prefer the authors to clearly explore this further. Given the population under study, I believe some additional assumptions would be valuable.
Author Response
Comments and Suggestions for Authors
This paper examines performance and participation in the Vasaloppet XC skiing race over almost a century. There are few previous publications on this subject, and those that exist are cited. The introduction and methods are appropriate and detailed enough to convey an understanding of the project. I have a couple of suggestions for grammer/spelling, which are minor, but also a concern with the relation to health. All of these should be easily addressed and I have detailed them as follows:
Line 128 Finish should be corrected to Finnish
Answer: We agree with the expert reviewer and corrected the spelling error.
Line 156 Finish should be corrected to Finnish
Answer: We agree with the expert reviewer and corrected the spelling error.
Lines 195 and 203 'Engadiner ski marathon' should be consistently changed to 'Engadin ski marathon'
Answer: We agree with the expert reviewer and corrected the spelling error.
Line 240 rises should be corrected to raises
Answer: We agree with the expert reviewer and corrected the spelling error.
Line 244 biathlets should be corrected to biathletes and russia should be Russia
Answer: We agree with the expert reviewer and corrected the spelling error.
Lines 79-83 suggest are a strong follow up statements supporting the logic of the investigation. However, given the results and discussion, very little attention was given to importance from a health perspective. I understand public policy and appreciate the attempt at acknowledging the lack of participation of females, but would prefer the authors to clearly explore this further. Given the population under study, I believe some additional assumptions would be valuable.
Answer: We agree with the expert reviewer and inserted the following section (line 90-94): Perspective. A large study comparing skiers in the XC race and non-skiers provided additional information for the hypothesis on individuals with high level of physical activity representing a healthy lifestyle, have a lower risk of CVD or death (U. Hallmarker Eur Heart J Qual Care Clin Outcomes. 2018 Apr 1;4(2):91-97; https://www.ncbi.nlm.nih.gov/pubmed/29390055).
Reviewer 3 Report
The study is well done and there is also merit in the entire gender discussion.
In general the study is well presented and nicely done.
The main conclusion is very shallow and should be revised. What is the development in the ratio m/w. Did the performances difference between m/w change? Where will it go? For this conclusion the entire work is not required. A simple glimps at the data is sufficient.
Author Response
Comments and Suggestions for Authors
The study is well done and there is also merit in the entire gender discussion.
In general the study is well presented and nicely done.
The main conclusion is very shallow and should be revised. What is the development in the ratio m/w. Did the performances difference between m/w change? Where will it go? For this conclusion the entire work is not required. A simple glimpse at the data is sufficient.
Answer: We agree with the expert reviewer and changed the Conclusion to: Based on these findings, we concluded that a relatively small number of women pariticipates in XC skiing. Nevertheless, it is noticable that over the years more participation of women in XC skiing increased, which might lead to a change of the men-to-women ratio over time. The increased participation of women already has an impact on their race time. Therefore, the development of public health policies targeting the participation of women in XC skiing should be a concern in the countries with tradition in this sport.
Round 2
Reviewer 1 Report
The article has been revised to an acceptable standard
Reviewer 3 Report
well done.